# Multi-state detection and spatial addressing in a microscope for ultracold molecules

Jonathan M. Mortlock ⬤ ✉, Adarsh P. Raghuram ⬤, Benjamin P. Maddox ⬤,
Philip D. Gregory ⬤ & Simon L. Cornish ⬤ ✉

Precise measurement of the particle number, spatial distribution and internal state is fundamental to all proposed experiments with ultracold molecules both in bulk gases and optical lattices. Here, we demonstrate in-situ detection of individual molecules in a bulk sample of $^{87}Rb^{133}Cs$ molecules. Extending techniques from atomic quantum gas microscopy, we pin the molecules in a deep two-dimensional optical lattice and, following dissociation, collect fluorescence from the constituent atoms using a high-numerical-aperture objective. This enables detection of individual molecules up to the resolution of the sub-micron lattice spacing. Our approach provides direct access to the density distribution of small samples of molecules, allowing us to obtain precise measurements of density-dependent collisional losses. Further, by mapping two internal states of the molecule to different atomic species, we demonstrate simultaneous detection of the position and rotational state of individual molecules. Finally, we implement local addressing of the sample using a focused beam to induce a spatially-dependent light shift on the rotational transitions of the molecules.

Ultracold polar molecules have many promising applications in quantum science[1–3], in part due to the possibility of engineering quantum gases with tunable, long-range and anisotropic dipole-dipole interactions. These dipole-dipole interactions are intimately connected with control of molecular rotational states[4]. The engineering of dipolar spin-exchange interactions in particular requires the preparation of molecules in mixtures or superpositions of different rotational states. Such interactions are a key component in current experimental realisations of spin-1/2 models of quantum magnetism with polar molecules[5–9]. For disordered samples, simultaneous read-out of the distribution of density and spin is crucial to accurately determine the state of the system, and therefore enable accurate comparison to theoretical models[10–12].

One of the leading approaches to preparing ultracold molecular gases generates relatively small but dense samples of molecules by association from pre-cooled atomic gases[2]. This approach has allowed the production of quantum-degenerate samples of fermionic KRb[13] and NaK[14], and bosonic NaCs[15]. In these experiments,

detection is typically performed by breaking the molecules apart and taking absorption images of the resulting atoms[16]. This technique has advantages; the imaging of alkali-metal atoms is sensitive and straightforward, and, as the dissociation process addresses a single hyperfine and rotational state of the molecule, the detection is inherently state sensitive. However, the dissociation process can add energy to the atoms, thereby affecting the spatial distribution, and the typical noise floor is of order ~100 molecules. Direct detection via absorption imaging of associated molecules has also been demonstrated[17]. However, the signal-to-noise for small samples is limited due to the molecules lacking sufficiently closed cycling transitions to scatter many photons. Ultracold molecules may also be detected via a combination of ionisation spectroscopy and velocity-map imaging[18]. This enables flexible detection of atomic and molecular species that can give unique insights into chemical processes[19], but requires a purpose-built apparatus and does not directly probe the spatial distribution of molecules in the sample.

Department of Physics, Durham University, South Road, Durham DH1 3LE, United Kingdom. ✉e-mail: jonathan.m.mortlock@durham.ac.uk; s.l.cornish@durham.ac.uk

In atomic systems, quantum gas microscopy[20,21] has enabled experiments that can detect down to a single atom, and has enabled access to a new range of local observables[22]. The technique uses a deep optical lattice to pin atoms while they fluoresce under illumination with cooling light. This fluorescence is collected with a high-resolution imaging system, and the lattice structure is used to deconvolve the image to resolve single atoms. This technique can be extended to allow simultaneous readout of atomic density and spin state[23], allowing the study of microscopic correlations of spin and charge[24]. These techniques are already being applied to experiments with arrays of ultracold molecules prepared in optical lattices. Pioneering work by Christakis and Rosenberg et al.[7,25] has demonstrated the resolution of correlations between pairs of molecules arising from their quantum statistics, and observed correlations from dipolar interactions between nearest-neighbours in the lattice. These experiments complement those performed on molecules in optical tweezer arrays[26,27] where a similar detection scheme based upon collecting atomic fluorescence is employed.

Recently, microscopy techniques have been extended beyond lattice-confined samples and applied to the study of bulk atomic gases. That is, the atoms are initially not confined to a lattice until a pinning lattice freezes the atom in position for detection. This effectively takes a snapshot of the positions of atoms in the sample at the resolution of the lattice spacing. This has enabled the precise study of correlations in the continuum for quantum degenerate or near-degenerate Bose and Fermi systems[28–32]. In our work, we extend these techniques further by applying them to bulk molecular gases.

Here we demonstrate spatially-resolved detection of single ultracold $^{87}$Rb$^{133}$Cs (RbCs) molecules in a thermal bulk gas. Our technique, summarised in Fig. 1, generates images of fluorescence captured from single atoms produced by breaking apart molecules that are pinned in place using a 2D optical lattice. We use these images to reconstruct the molecular density in the trap, enabling precise tracking of one- and two-body losses. We also show that we can detect molecules in two different rotational states in a single experimental cycle. This is achieved by mapping the rotational states onto the atomic species left in the lattice site for detection. Finally, we add a local addressing beam. This allows us to transfer a chosen area of the sample into a different rotational state. This provides an additional test of the spin and spatial resolution of our imaging technique. We observe and analyse the time-of-flight expansion of the selected area as a novel approach to thermometry of the molecules that are captured in the lattice.

## Results

### Detection of single molecules

Our sample initially consists of ~1500 RbCs molecules confined to an optical dipole trap; see Methods for details of the preparation sequence. The dipole trap is formed at the intersection of a light sheet, with a tight vertical waist of 7 μm, and a pair of circular beams, with all three beams propagating in the horizontal $xy$ plane (the coordinate systems is defined in Fig. 1a). We calculate the trap frequencies experienced by molecules in the rotational ground state ($N = 0$) to be $(\nu_x, \nu_y, \nu_z) = (100, 200, 680)$ Hz.

An overview of our detection scheme is shown in Fig. 1b. The first step is to pin the molecules in place using a 2D optical lattice, thereby preserving spatial information about the gas. The lattice is formed from a single $\lambda = 1064$ nm beam that is retro-reflected in a bow-tie configuration, as illustrated in the lower part of Fig. 1a, with 1/$e^2$ waists of 100 μm at the location of the molecules. This geometry yields a square lattice with spacing $a_{lat} = \lambda/\sqrt{2} = 752$ nm and axes that are rotated by 45° with respect to $x$ and $y$[33]. To pin the molecules, we ramp on the 2D optical lattice in 0.1 ms to a depth of $7.0(1) \times 10^3 E_{rec,RbCs}$ (expressed in units of lattice recoil energy $E_{rec} = h^2/8ma_{lat}^2$, where $m$ is the mass of RbCs molecules) whilst simultaneously increasing the light sheet power to provide tight confinement vertically.

We then break apart the RbCs molecules into their constituent Rb and Cs atoms. The dissociation process is the reverse of the magneto-association and stimulated Raman adiabatic passage (STIRAP) steps used to form the molecules[34,35], and converts each molecule into a Rb atom in $5S_{1/2}$ ($F = 1$, $m_F = 1$) and a Cs atom in $6S_{1/2}$ ($F = 3$, $m_F = 3$). As the efficiency of STIRAP is reduced when the trap light intensity is high[36], we lower the lattice intensity to $< 120\ E_{rec,RbCs}$ for 0.1 ms to perform the return STIRAP to the Feshbach state. With this approach, we achieve a transfer fidelity of 95(1)%, comparable to efficiencies achieved in similar experiments where STIRAP is performed in free space[36]. There is scope to increase this fidelity using feed-forward suppression of laser phase noise[37]. The time for which the lattice intensity is reduced is short compared to the expected time for trapped molecules to tunnel between sites (>5 ms), although some molecules may occupy highly-excited motional states in the lattice and, therefore, may be relatively free to move. However, we estimate that this leads to minimal blurring of the spatial information, on the order of one or two sites at most. To avoid light-assisted collisions between the Rb and Cs atoms during the imaging step, we selectively remove

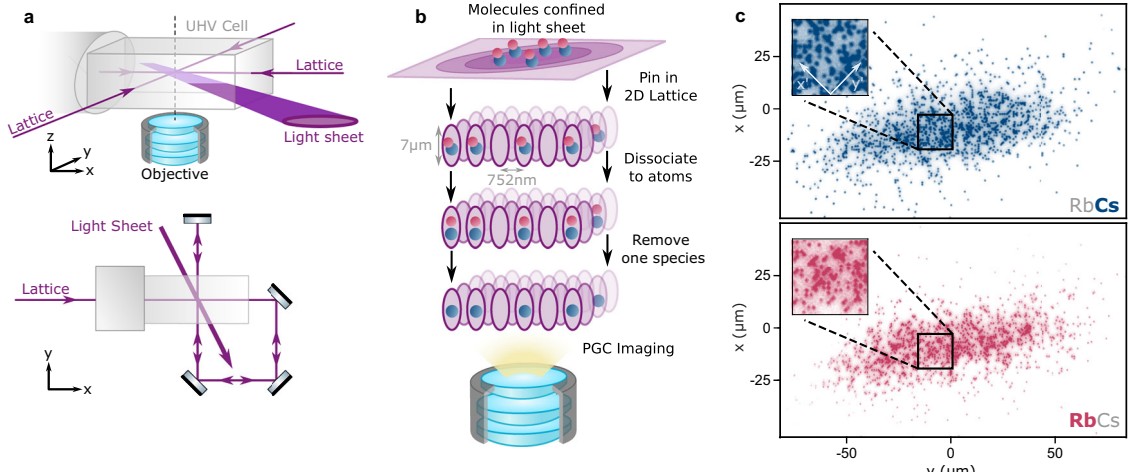

**Fig. 1 | Detection of single molecules using an optical lattice. a** Overview of the experimental setup and **b** our protocol for detecting single molecules based on pinning the molecules in place and imaging single atoms after dissociation, as explained in the main text. We define the lattice beam axes to be $x$ and $y$, and the vertical direction $z$. **c** Typical fluorescence images at the peak molecule density used, with single atoms clearly resolved. Lattice axes ($x'$ and $y'$) are indicated in the inset. We image either Rb or Cs atoms from each RbCs sample, hence these are images from different realisations of the experiment.

one of the species from all sites after the dissociation. This is achieved by a short pulse of resonant light, as described in "Methods".

To perform imaging, we make the atoms fluoresce using polarisation gradient cooling (PGC). This allows us to excite fluorescence while maintaining a low enough temperature that the atoms remain pinned in their lattice sites[20,21]. Both atomic species experience a trap depth > 4000 $E_{rec}$ in the pinning lattice. The fluorescence is collected using an objective lens with a numerical aperture of 0.7 positioned below the sample and imaged onto a CMOS camera. We typically collect fluorescence for 750 ms to clearly resolve the signal from individual atoms above the background noise. In "Methods" we present estimates of the detection fidelity, signal-to-noise ratio, and the point-spread function of the imaging system. Example images where we choose to keep either the dissociated Rb or Cs atoms are shown in Fig. 1c. Using a deconvolution algorithm based on a single-layer neural network[38], we are able to reconstruct the lattice occupancy and hence determine the spatial distribution of molecules. We estimate the fidelity of site reconstruction is limited by the deconvolution process to around 95% for our most dense molecular samples (see "Methods" and Supplementary Fig. 1).

We observe a binary fluorescence histogram associated with the occupation of each lattice site, with the two peaks indicating the presence of 1 or 0 molecules (see Supplementary Fig. 1). This is because our detection scheme leads to the loss of pairs of molecules that occupy the same site of the pinning lattice. This pair loss likely occurs when the molecules are first loaded into the lattice; RbCs molecules are known to undergo rapid loss at close to the universal rate due to optical excitation of two-body collision complexes[39,40]. Additionally, pair loss may occur during the imaging step due to homonuclear light-assisted collisions, as observed in atomic gas microscope experiments[22]. These effects combine to lead to the detection of a single molecule on sites that initially contain an odd number of molecules, while sites that initially contain an even number of molecules will appear empty. We note, however, that in all of our experiments, the molecule density is sufficiently low that there is a negligible probability of more than two molecules occupying the same site of the pinning lattice.

## Reconstruction of the molecular density

Our technique effectively captures the molecule distribution at the time that the pinning lattice is ramped on, thus enabling a direct measurement of the molecular density in the trap. For our densest samples, however, care must be taken to properly account for the pair loss to accurately determine the true molecular density distribution.

We correct the observed density of filled sites to reconstruct the true density profile. We first extract the average observed site occupancy $n'_{site}$ by spatially averaging across our images with a square ($7 \times 7$ site) uniform kernel. The kernel size is chosen to be small compared to the observed variations in molecule density within the trap. The molecule sample is thermally distributed and at equilibrium prior to the pinning, such that we expect no strong density correlations on the scale of the lattice spacing. To find the corrected site occupancy $n_{site}$, we assume that the number of molecules pinned to each site is Poisson distributed. Thus, the pair loss projection converts $n_{site}$ to $n'_{site}$ via the relation $n'_{site} = \frac{1}{2}(1 - e^{-2n_{site}})$, plotted in Fig. 2a. By inverting this relationship, we determine the local density to be $n_{site}(\mathbf{r}) = -\frac{1}{2}ln(1 - 2n'_{site}(\mathbf{r}))$, which we then convert to a molecule column density using the known lattice spacing. At this stage, we also correct for the finite STIRAP efficiency. The stages of this process are depicted in Fig. 2b. This approach is valid for our experiments, as the peak occupancy is generally low, $n'_{site} \leq 0.3$.

Measurement of the in situ column-density distribution, when combined with the known trap frequencies, allows full determination of the thermodynamic properties of the gas. We fit a 2D Gaussian function to this distribution to find thermal $1/e$ widths of the cloud,

$\sigma_a = 8(1)$ μm and $\sigma_b = 30(1)$ μm, where the directions are defined by the major and minor axes of the observed cloud as shown in Fig. 2b. Given the known trap frequencies, these widths correspond to a sample temperature of 2.3(4) μK. Using this temperature and the vertical trap frequency, we estimate the width along the unseen vertical direction to be $\sigma_v = 5.6(8)$ μm and convert the peak column density to a mean density via the factor $4\sqrt{\pi}\sigma_v$.

We apply our technique to study collisional losses in the thermal gas of molecules. Here, the density dependence of the loss is critically important to verify the mechanisms at play. Previous studies of collisions in RbCs have relied upon detection of the molecule number via absorption imaging, which is then converted to the density through independently measured temperatures and trap frequencies. Our approach, however, yields a more direct measurement of the density, in addition to significantly enhanced precision when measuring small numbers of molecules. In Fig. 2c, we plot the mean density of the sample as a function of hold time in the dipole trap, with the longest hold times and therefore lowest densities corresponding to the detection of approximately 50 molecules. We fit the results using the solution to the rate equation $\dot{n}(\mathbf{r},t) = -\sum_m k_m n(\mathbf{r},t)^m$, where $m = 1, 2$ corresponding to one- and two-body losses, respectively. We find rate coefficients of $k_1 = 0.49(2)$ s$^{-1}$ and $k_2 = 5.1(4) \times 10^{-11}$ cm$^3$ s$^{-1}$. Given the relatively high trap intensity of 6 kW cm$^{-2}$, we expect to see a one-body loss rate of 0.28 s$^{-1}$ due to photon scattering, based upon measurements of loss performed in optical tweezers at a wavelength within 1 nm of the value used here[27]. Our observed one-body rate coefficient is within a factor of 2 of this rate; the difference may be attributed to the presence of two-photon transitions to more highly excited electronic states[41] that cause strong wavelength dependence around 1064 nm. The two-body rate coefficient we measure is a factor of two lower than the thermally-averaged universal rate, and agrees well with the rates previously measured in bulk gases[39].

By choosing not to remove excess Rb atoms after formation of the molecules, we can investigate atom-molecule collisions, as shown in blue in Fig. 2c. Using absorption imaging, we measure a constant Rb number of $2.1(1) \times 10^4$ over a hold time of 0.5 s. This corresponds to a mean atomic density of $8.6(5) \times 10^{11}$ cm$^{-3}$. Using our technique, we measure rapid loss of RbCs immersed in this gas, and measure a decay rate of $k_1 = 13(1)$ s$^{-1}$. From this, we extract an atom-molecule collision rate of $1.5(5) \times 10^{-11}$ cm$^3$ s$^{-1}$, in good agreement with previous measurements[42].

## Multi-state detection

We extend our imaging technique to perform simultaneous two-state readout of the molecules. We project the rotational state of the molecule onto the species of atom remaining in the image, thereby enabling detection of the populations of two rotational states in each iteration of the experiment. Mapping the rotational states of the molecule onto the spin-1/2 {↓, ↑} basis, our method allows complete characterisation of a disordered spin system in a single experimental instance. Our approach is similar to that proposed in[11] and complements the spin read-out techniques developed for molecules in optical tweezer arrays[27,43]. The process relies on the state specificity of the reverse STIRAP and dissociation sequence, which only converts molecules in a single hyperfine level of the rotational ground state into atom pairs, and on the ability to selectively remove atoms based on the hyperfine state that they occupy.

We choose to use the rotational states $|\downarrow\rangle = (N = 0, M_F = 5)$ and $|\uparrow\rangle = (1, 6)$. Here, $M_F = M_N + m_{Rb} + m_{Cs}$ is the projection of the total angular momentum along the quantisation axis (defined by a 181.6 G magnetic field oriented vertically along $z$), $M_N$ is the projection of the rotational angular momentum, and $m_{Rb}$ ($m_{Cs}$) is the projection of the nuclear spin of Rb (Cs). At this magnetic field, $N$ and $M_F$ are generally the only good quantum numbers[44]. However, our chosen states are also spin stretched such that they also correspond to the uncoupled

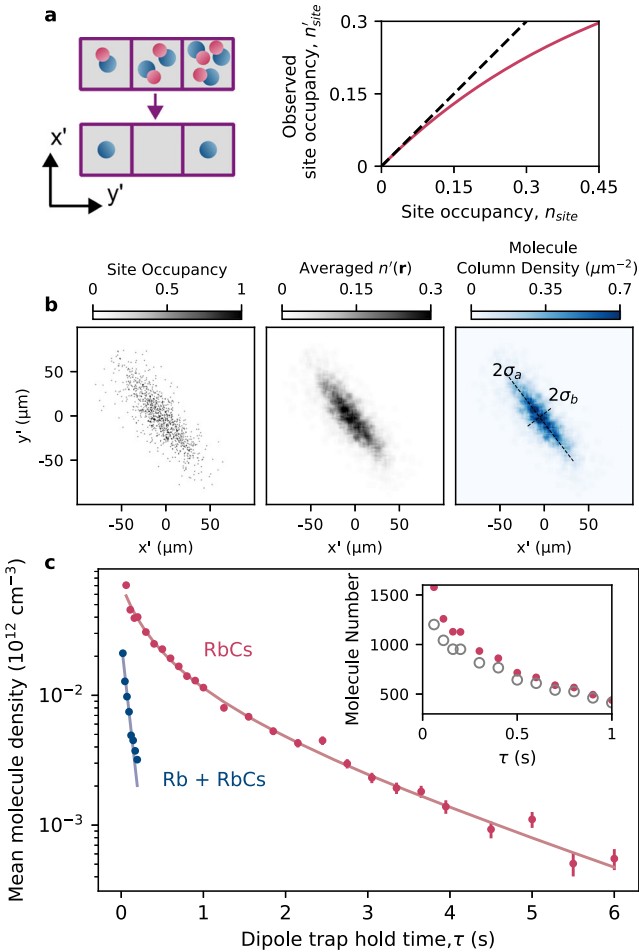

**Fig. 2 | Measurement of the molecular density. a** For lattice sites with multiple molecules, we expect pair loss, as indicated in the cartoon, leading to the observed site occupancy being lower than the actual site occupancy. Assuming a thermal distribution, such parity projection losses can be calculated, leading to the solid red line shown. **b** Method for determining molecule density. Inverting the relationship shown in a allows us to correct for pairwise losses in imaging. Panels show the progression from site-resolved images of molecules, which are spatially averaged to determine the mean site occupancy and finally the molecule column-density distribution. Fitting to this data gives the peak column density and cloud widths ($\sigma_a, \sigma_b$). **c** We determine the mean molecule density and plot this as a function of hold time to study collisional loss in the dipole trap, with and without Rb atoms, error bars represent the propagated uncertainty due to molecule shot noise. Inset shows the difference between the corrected molecule number (solid circles) and the uncorrected number of observed atoms (open circles), which becomes small for molecule numbers below 500. We fit the data with a loss model (solid lines) to determine density-dependent loss coefficients.

basis states $|\downarrow\rangle = (N=0, M_N=0, m_{Rb}=3/2, m_{Cs}=7/2)$ and $|\uparrow\rangle = (1, 1, 3/2, 7/2)$. Crucially, the $|\downarrow\rangle$ state is directly addressed with STIRAP, while the $|\uparrow\rangle$ state is not. We can coherently drive the population between $|\downarrow\rangle$ and $|\uparrow\rangle$ using resonant microwaves.

The sequence of operations for spin projection is illustrated in Fig. 3a. In the first step, $|\downarrow\rangle$ molecules are dissociated to atom pairs, while $|\uparrow\rangle$ molecules are off-resonant with the STIRAP and are not transferred to the Feshbach state. A microwave $\pi$-pulse is then applied to return the $|\uparrow\rangle$ molecules to the recoverable $N=0$ state. After this, the Rb atom from the $|\downarrow\rangle$ state is transferred from ($F=1, m_F=1$) to the uppermost hyperfine state (2, 2) using microwave adiabatic rapid passage (ARP) and the Cs atom is then removed. The $|\uparrow\rangle$ molecules are then dissociated, and a second Rb ARP is performed. This puts the Rb atom from $|\uparrow\rangle$ in (2, 2) and simultaneously returns the Rb atom from

$|\downarrow\rangle$ molecules back to the lower hyperfine state (1, 1). We then selectively remove Rb atoms in (2, 2) associated with $|\uparrow\rangle$ molecules using light on the cycling transition $5S_{1/2}(2, 2) \rightarrow 5P_{3/2}(3, 3)$. At the end of this process, the sites that initially contained a molecule $|\downarrow\rangle$ are mapped to an Rb atom, and those that contained a molecule $|\uparrow\rangle$ are mapped to a Cs atom.

Detection of both species in a single experimental cycle is performed by a series of sequential images of fluorescence. Thanks to the large separation of the imaging wavelengths, we find that the atoms are unaffected by imaging the other species. Thus, to reliably determine the relative locations of the atoms, we take three successive images as shown in Fig. 3b, collecting fluorescence first from Rb only, then Cs only, and finally Rb and Cs together. The final image removes any ambiguity in the position of the lattice between different images. This allows us to reconstruct a complete picture of the spin density as shown.

In Fig. 3c, we demonstrate our spin detection technique using a simple Rabi oscillation experiment. We drive the $\sigma^+$ transition between $|\downarrow\rangle$ and $|\uparrow\rangle$ by applying resonant microwaves for a variable duration and use the multi-state readout to detect both rotational states. Here, the trap light is switched off during the microwave pulse to avoid light shifts of the transition[41]. From the fits to the measurement, we determine a Rabi frequency of 20.0(1) kHz, which is low enough to avoid off-resonant coupling to other hyperfine states due to imperfect microwave polarisation. Since the microwave drive is performed in free space, thermal expansion of the cloud leads to loss of molecules when recaptured by the pinning potential. We model this using an exponential decay and fit a 1/e time around 700 µs. The average initial molecule number detected via the two spin-mapping channels agrees within the statistical uncertainty; fits to the Rabi oscillations yield $2.61(16) \times 10^2$ molecules from Rb and $2.58(11) \times 10^2$ molecules from Cs. From independent measurements characterising each stage of the multi-state detection sequence, we expect the overall probability of faithfully mapping the state of an isolated molecule to the correct atom to be around 84(2)% for both states. This is currently limited by technical factors, such as the ARP fidelity and STIRAP efficiency, as discussed in the Methods. Based on the best reported measurements for STIRAP in RbCs[37] and state-of-the-art transfer fidelities between atomic hyperfine states[45], we expect that a mapping fidelity > 98% is possible.

## Spatially resolved addressing of molecules

We also demonstrate local addressing of the rotational transition in a selected region of the molecular gas. This is achieved by illuminating part of the sample to light shift the molecular transition off resonance, as illustrated in Fig. 4a. Here, we address the molecules with light at 817 nm, where the anisotropic polarisability is large and negative[27,46]. This produces a strong light shift of the transitions from $N=0$ to the hyperfine states of $N=1$[47] as shown in Fig. 4b. We then apply a microwave $\pi$-pulse on the transition from $|\downarrow\rangle$ to $|\uparrow\rangle$. In regions where the light shift is greater than the Fourier width of the microwave pulse, molecules will remain in $|\downarrow\rangle$, while molecules outside of this region will be excited to $|\uparrow\rangle$. The boundaries of the addressed region form a circle with a 12.5 µm radius, indicated by a dashed circle in the exemplary spin-resolved images in Fig. 4c, d. From the observed gradient of the addressing beam intensity at this radius, we expect a blurring of the edge of this circle on the order of 5 µm.

Spatial addressing also allows us to selectively remove molecules from the sample. For example, we can remove all molecules outside of the addressing beam by applying the local-addressing pulse, pinning the molecules, and then discarding those in $|\uparrow\rangle$. This final step is achieved by performing a global $\pi$ pulse on the $|\uparrow\rangle \leftrightarrow |\downarrow\rangle$ transition with the addressing beam switched off, thus exchanging the state occupied by molecules in the two regions, followed by dissociation and resonant removal of the atoms in $|\downarrow\rangle$ that now occupy the area outside the addressing region.

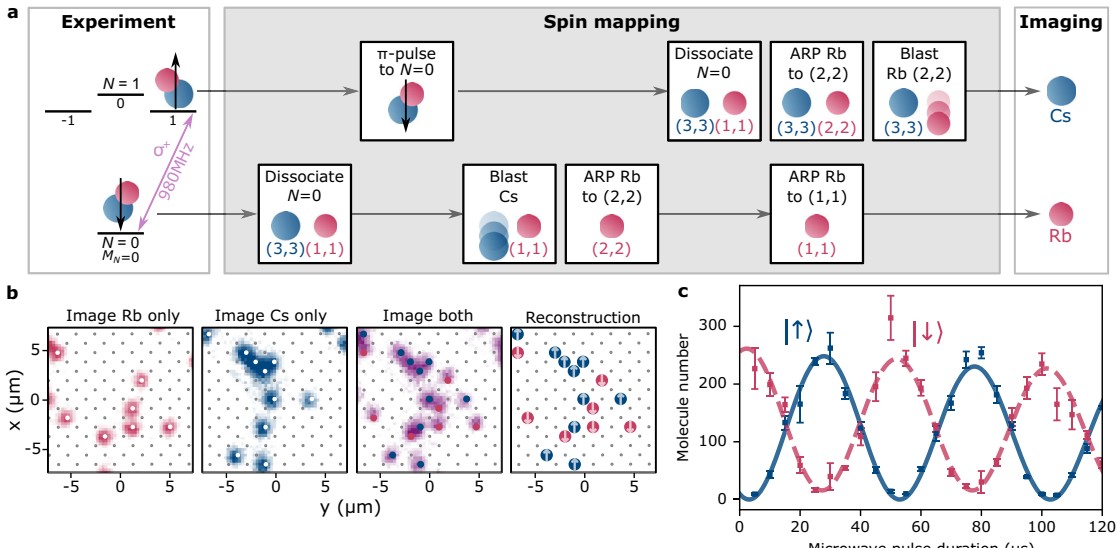

**Fig. 3 | Simultaneous two-state detection of molecules. a** Overview of the sequence used to map the internal state of the molecule to the recovered atomic species. The molecule states are labelled by rotational numbers $N$, $M_N$ and the atoms by hyperfine ($F$, $M$). **b** At the end of each experiment, we take three successive images as shown to allow us to reconstruct the spin distribution. The first two images are taken of each species alone, where grey dots show the lattice sites and white dots represent the detected occupancy after deconvolution. The third image, of both species simultaneously, confirms the relative spatial distribution of the Rb (red dots) and Cs (blue dots). The spin distribution can then be reconstructed with spins indicated by white arrows. **c** We measure a Rabi oscillation on the molecular transition to characterise our simultaneous readout. Error bars represent the standard error in each data point.

We can measure the thermal energy distribution of the remaining subset of pinned molecules by ramping off the pinning potential and observing expansion of the sample as a function of the time of flight, as shown in Fig. 4e. To detect the expansion, we re-pin the cloud after a variable time of flight and image as before. Each realisation of this sequence produces ~100 molecules. Building up 2D histograms from many shots allows us to measure the expansion of the cloud. Using a quadratic fit to the results, we determine the temperature to be 5.0(1) μK. The temperature observed here is higher than for the initial molecule sample due to the non-adiabatic loading and unloading of the pinning lattice.

In a complementary measurement, we quantify the energy distribution of the initially pinned molecules by examining the movement of molecules between sites as a function of the lattice depth. Starting from the localised sample, we ramp the lattice down to a variable depth and then observe the mean radius of the molecules from the centre of the addressing beam after 1 ms. We see that for lattice depths below 1000 $E_{rec}$ the radius grows over time, indicating that the molecules are no longer pinned, as shown in Fig. 4c. We compare these results to a Monte Carlo model (see "Methods") where we assume that molecules maintain the band index in the lattice when initially pinned. As the lattice depth is decreased, the number of bands below the trap depth decreases, and the highest energy molecules freely diffuse around the lattice. We model this as thermal expansion with a temperature corresponding to that measured in Fig. 4e. We find good quantitative agreement between this model and our observations. We also note that for lattice depths above 1000 $E_{rec}$ the molecule cloud radius does not significantly increase, and we measure no change in the cloud size at long times up to 1000 ms; an example image at this long hold time is shown in the inset of Fig. 4d.

## Discussion

We have applied microscopy techniques to bulk molecular gases produced via association. By pinning the molecules in place prior to dissociation, we detect both the position and the state of molecules in the gas with single-molecule sensitivity. We have shown how to accurately reconstruct the density distribution of the original molecular gas

and used this approach to precisely measure the density-dependent rate of loss of molecules from the trap. We have implemented simultaneous detection of molecules occupying two different rotational states by mapping the molecular state onto the atomic species left in the pinning lattice after dissociation. Finally, we have shown that we can address selected regions of the gas, allowing the preparation of spatially separated spin regions or the removal of all but a chosen subset of the molecules. We expect that our detection methods may be broadly applied to the full range of bialkali molecules that are produced by association, given that atomic quantum gas microscopy has been applied to every stable species of alkali atom.

Our work has immediate applications in the study of ultracold molecular collisions, where many fundamental questions remain to be addressed[48]. For example, it has been predicted that the rate of collisional loss can vary greatly for pairs of molecules prepared in different rotational states[49]. This can be tested by preparing a small impurity of molecules in a larger bath of molecules occupying a different state, such that the collisional loss of impurity molecules is dominated by collisions with the bath. Our technique allows small numbers of molecules to be accurately resolved, and the density profile associated with the impurity and the bath to be measured simultaneously. In a similar vein, our methods can be used to study atom-molecule collisions, as we have demonstrated, enabling sensitive searches for collision resonances[50], for example.

A major goal in the field of ultracold molecules is to study many-body physics using dipole-dipole interactions between molecules held in optical lattices[1]. Here, we demonstrate all the key components required for multi-state microscopic detection and spatial addressing in such a system. Currently, the high temperature of our molecules after pinning precludes such a study, as the large number of occupied motional states will lead to both strong differential light shifts and rapid dephasing of interactions. However, these problems can be overcome by using a magic wavelength[8] optical lattice and preparing the molecules in the ground band following the established protocol for RbCs[51,52]. Our spin-resolved approach is also ideally suited to studies of spin-squeezing arising from the dipolar interaction[53,54].

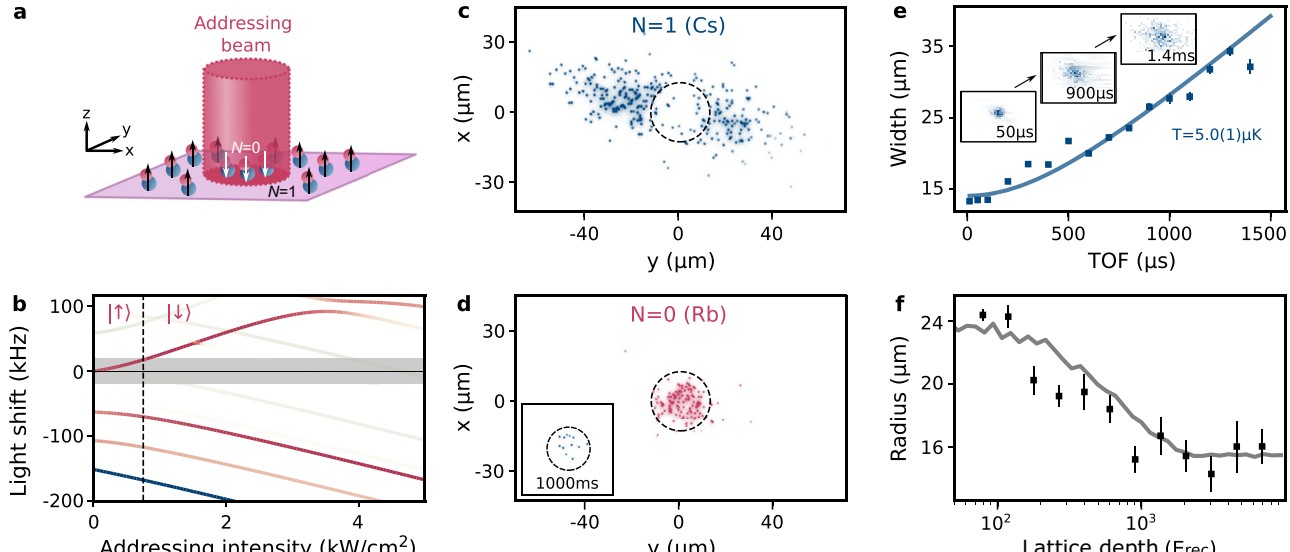

**Fig. 4 | Spatially-resolved addressing of the rotational state. a** The addressing beam is focused onto the cloud to induce a local light shift of the $N=0$ ($|\downarrow\rangle$) to $N=1$ ($|\uparrow\rangle$) rotational transition in the molecule. **b** The light shift as a function of addressing beam intensity. Line colours indicate the relative strengths of transitions for microwaves polarised along $z$ and in the $xy$ plane as in[47], and the grey shaded band indicates the Fourier width of the $\pi$-pulse used to transfer molecules from $N=0$ to $N=1$. The vertical black dashed line shows the threshold intensity for shielding the molecules from excitation. **c**, **d** Example spin-resolved images following a $\pi$-pulse on the free-space rotational transition with the addressing beam on. The dashed circles indicate the extent of the addressed region. The inset in (**d**)

shows addressed molecules held in the lattice at the highest available depth for 1000 ms. **e** We produce a small localised sample by pinning the molecules in a deep lattice, and then selectively removing the unaddressed molecules. We then ramp off the lattice and measure the width of the cloud after a variable time of flight (TOF) in free space. Example images are shown inset. Error bars represent the uncertainty in the fitted cloud radius. **f** Using a similar protocol to prepare a small sample, we measure the radius after a fixed hold time of 1 ms as a function of the lattice depth. The grey line shows a Monte-Carlo simulation of the evolution. Error bars represent the standard deviation in the experimental data.

## Methods

### Production of ultracold RbCs molecules

The experimental apparatus used here has been described in previous works[55,56], though it is distinct from that used for our previous work with RbCs molecules in bulk gases. We utilise a multi-chamber vacuum design, where the atoms are first trapped and cooled in a stainless steel chamber and then transported to a rectangular glass cell using an optical conveyor-belt[56]. Our process for forming the ultracold mixture is illustrated in Supplementary Fig. 2. The two species are prepared sequentially and are loaded into two spatially separated optical dipole traps in the cell. The Rb atoms are prepared first, transported into the cell and then moved by 1 mm using a movable optical trapping beam. Cs is then cooled, transported and loaded into a separate fixed-position dipole trap. With the species separated, we are able to evaporatively cool both species simultaneously and prepare Bose-Einstein condensates of $1 \times 10^5$ Rb atoms and $4 \times 10^4$ Cs atoms.

To form molecules, we merge the two atomic gases to form a mixture with high phase-space density (PSD). Due to the immiscibility of the atomic condensates at magnetic fields where Cs is stable against three-body losses[57], we form molecules from a thermal mixture. We use the Moving Dimple beam shown in Supplementary Fig. 2 to merge the traps when the atoms are just above their respective condensation temperatures. This generates a mixture containing ~$2 \times 10^5$ of each species at a PSD of ~0.1 and a temperature of ~200 nK.

Our scheme for molecular association follows that used in[34,35]. We first associate molecules into the weakly-bound Feshbach state by ramping the magnetic field over the Feshbach resonance at 197 G, continuing the ramp to 181.6 G to access a more deeply bound Feshbach state. We then transfer the molecules to their rovibrational ground state using STIRAP and remove any remaining atoms using resonant light (as described below). As mentioned above, for the measurements here, we observe a one-way STIRAP transfer fidelity of $\eta_{\text{STIRAP}} = 95(1)\%$

### Optical trap light

All of the light used to trap the molecules is derived from the same source operating at $\lambda = 1064.521(4)$ nm, with different beams detuned by ~10 MHz from each other so that interferences are time averaged on the scale of the atomic or molecular trapping frequencies. The dipole trap, which holds the ground state molecules before they are pinned, is formed by a light sheet beam with vertical waist 7 μm and horizontal waist 150 μm, and two circular beams of waists 45 μm and 100 μm.

### Atom removal with resonant light

To selectively remove one of the atomic species from the pinning lattice at our operating magnetic field of 181.6 G, we use circularly polarised resonant light propagating along the quantisation axis. For spin-resolved detection, the Rb light is tuned to address the Zeeman-shifted transitions $5S_{1/2}(F=2, m_F=2) \rightarrow 5P_{3/2}(3, 3)$. For detection not requiring mapping of the internal state onto the atomic species, an electro-optic modulator (EOM) is used to generate additional light to address the $5S_{1/2}(1, 1) \rightarrow 5P_{3/2}(2, 2)$ transition. This allows the removal of Rb atoms in the hyperfine ground state that is occupied after dissociation without the need for ARP. The Cs removal light also uses an EOM; the carrier is resonant with $6S_{1/2}(4, 4) \rightarrow 6P_{3/2}(5, 5)$ and one of the sidebands is tuned to the $6S_{1/2}(3, 3) \rightarrow 6P_{3/2}(4, 4)$ transition.

When using the EOM for both species, we observe the mean atom number remaining after the removal pulse to be 1.9 for Rb and 1.8 for Cs, when starting from initial numbers ~$1 \times 10^5$. The $1/e$ lifetime during the removal pulse is measured to be around 5 μs. This is much faster than the expected mean time between collisions of ~1 ms, such that pair loss caused by the removal of light will be unlikely. If we omit these removal pulses, we observe pair loss during simultaneous PGC with a $1/e$ lifetime of on the order of 10 ms.

## Polarisation gradient cooling (PGC)

For both species, we perform PGC on the atomic $D_2$ lines using a single retro-reflected beam path (beam waist 0.5 mm), angled out of the $xy$ plane by 8 degrees, and rotated by approximately 10 degrees in the $xy$ plane from the direction of the lattice beam that defines the $y$ direction. The polarisation of the PGC light is arranged to give a $\sigma^+ - \sigma^-$ cooling configuration. We use 500 μW of cooling light for each species. This light is red detuned from the free-space cycling transitions by around 10 $\Gamma$, where $\Gamma$ is the natural line width of the relevant atomic transition. Repump light, addressing the lower hyperfine state of each atom, is provided by a separate beam propagating in the $xy$ plane. During the PGC, the magnetic field must be nulled, requiring a delay of around 15 ms for eddy currents from the high field coils to dissipate before beginning cooling. To collect fluorescence from both species during the same sequence, we either illuminate the atoms sequentially or simultaneously cool the atoms and use a filter wheel to isolate the fluorescence from each species. We find neither the hold time in the pinning lattice nor the presence of the cooling light for the other species leads to a significant loss of atoms.

## Characterisation of the atom detection

To establish the fidelity and signal-to-noise ratio (SNR) of our atom detection, we analyse images of dilute clouds[58]. We compare the detected signal from summing counts in $8 \times 8$ pixel regions of interest around isolated atoms with background from identically sized regions that are centred on empty parts of the lattice. An exemplary sparse image, and the resulting histograms are shown in Supplementary Fig. 1a–c. We define SNR $= (\mu_{atom} - \mu_{bg})/(\sigma_{atom} + \sigma_{bg})$ where $\mu_{atom,bg}$ is the mean and $\sigma_{atom,bg}$ is the standard deviation of the count distribution for atoms and background, respectively[58]. We find $SNR_{Rb} = 4.6$ and $SNR_{Cs} = 5.5$ for exposure times of 1 s. By comparing successive images of the same cloud, we determine the mean time for an initially pinned atom to be lost or move to a different site to be >10 s.

We determine the point-spread functions (PSFs) associated with our imaging system by averaging over the signals from 100 isolated Rb and Cs atoms. The PSF Airy radius ($r_{Airy}$) is found to be 1.09(12) μm for Rb and 1.04(10) μm for Cs. While the lattice spacing is below the Sparrow limit of our imaging system (0.77 $r_{Airy}$)[59], we are able to reconstruct the atomic filling at higher densities by deconvolving the images using a single-layer neural network[38]. We determine the lattice vectors from analysis of sparsely filled images, and assume these do not vary shot to shot. For each image, the lattice phase is determined by maximising the variance of the pixels closest to each lattice site, and for each site, a $20 \times 20$ pixel region centred on the site is fed into the neural network for reconstruction. The single-layer network is trained on simulated images and can be understood as performing a weighted sum of the counts on each site. This weighted sum is compared to a threshold to determine the site occupancy. Supplementary Fig. 1d shows the learned weights we use. To minimise the effect of variations in peak fluorescence, the images are normalised before being processed. A zoomed sample of the reconstructed data is shown in Supplementary Fig. 1e. We estimate the fidelity of detection by performing a double Gaussian fit to the neural network output, as shown in Supplementary Fig. 1f, which shows the histogram for the data at the highest density in Fig. 2. From this, we estimate an error rate of less than 5%.

## Internal state control

We use a pair of antennas, placed just outside the cell, to drive microwave transitions in the atoms and molecules. We drive the molecule transition between the $(N = 0, m_F = 5)$ and $(1, 6)$ states at 181.6 G, at a frequency of 980.384 MHz[44]. As we do not have control of the microwave polarisation, we use a Rabi frequency that is low enough to avoid off-resonant coupling to other hyperfine states. The Rb hyperfine state is controlled using adiabatic rapid passage (ARP) to transfer between $5S_{1/2}(F = 1, m_F = 1)$ and $5S_{1/2}(2, 2)$.

## Addressing light

The addressing light is formed by an 817 nm beam focused onto the molecules from above the cell with a $1/e^2$ waist of 14.6 μm and a peak intensity of $I = 3.3$ kW cm$^{-2}$. The light shifts plotted in Fig. 4 are calculated using methods from[60] and a previous measurement of the anisotropic polarisability at 817 nm of $\alpha^{(2)} = -2814(12) \times 4\pi\epsilon_0 a_0^3$ [27].

## Fidelity of the spin-mapping process

Here, we consider the technical factors that affect the fidelity of mapping the internal state of an isolated molecule in the pinning lattice to the correct tagging atom. The spin mapping protocol requires control of relatively large magnetic fields to dissociate the molecules, combined with a low-noise magnetic field environment for the microwave ARP transfer between the atomic states. Additionally, the part of the sequence where molecules are pinned should be performed as quickly as possible to avoid spontaneous Raman scattering of molecules from the intense lattice light.

In the work presented here, we developed two approaches. The first is to transiently turn off the high magnetic field to perform ARP at a low field where the magnetic field noise is significantly reduced. Using this approach, we measure ARP fidelities of 98(1)%. However, we need to introduce a time delay of 32 ms, largely limited by the time required for the field to stabilise when turning back on. This leads to loss of the $|\uparrow\rangle$ molecules, which must be held in the lattice for this time. We measure a survival probability of $\eta_{Hold} = 86(10)\%$ for these molecules, which strongly biases the detection fidelity in favour of the $|\downarrow\rangle$ molecules. An additional source of loss for the $N = 1$ molecules is off-resonant scattering from the Stokes light during the STIRAP transfer used to break apart the $N = 0$ molecules[27]. For this process, we measure a survival probability of $\eta_{Stokes} = 97(1)\%$. Taking all these factors together, yields transfer fidelities of $\eta_{|\downarrow\rangle} = \eta_{ARP}^2 \eta_{STIRAP} = 91(2)\%$ for $|\downarrow\rangle$ and $\eta_{|\uparrow\rangle} = \eta_{ARP}\eta_{hold}\eta_{Stokes}\eta_{STIRAP} = 78(9)\%$ for $|\uparrow\rangle$. We applied this approach to the data shown in Fig. 4.

The second approach involves performing ARP at high field, prioritising the speed of the sequence. Here measure a lower ARP fidelity of $\eta_{ARP} = 94(1)\%$, but have an increased survival probability of the pinned molecules of $\eta_{Hold} = 97(1)\%$ due to the significantly shorter 8 ms hold time. Thus, in this second approach, the transfer fidelities are $\eta_{|\downarrow\rangle} = \eta_{ARP}^2 \eta_{STIRAP} = 84(2)\%$ for $|\downarrow\rangle$ and $\eta_{|\uparrow\rangle} = \eta_{ARP}\eta_{hold}\eta_{Stokes}\eta_{STIRAP} = 84(2)\%$ for $|\uparrow\rangle$, as given in the Results. We applied this approach to the data shown in Fig. 3.

## Model for molecule motion in the lattice

To model this evolution, we map the thermal distribution of the molecules onto the bands of the optical lattice at full strength, assuming a temperature of 5.0(1) μK as measured in Fig. 4e. The band occupation is assumed to remain unchanged as the lattice adiabatically ramps down to a lower depth. The band energies are calculated for the lower lattice depth. Molecules in bands with energies higher than the trap depth become untrapped and expand ballistically with a velocity corresponding to their band energy. For each iteration, a molecule is initialised at a random position within the circle of the addressing beam and in a random lattice band sampled from the thermal distribution. The final position of the molecule following the 1 ms hold is then computed. The simulation is run for 10,000 iterations at each lattice depth. We also account for the residual unaddressed molecules that exist outside the addressing beam radius due to $\pi$-pulse imperfections, by allowing some probability that the molecule is initialised in an ellipse defined by major and minor radii extracted from a 2D Gaussian fit of the full cloud, as shown in Fig. 2b.

## Data availability

All data presented in this work are available in the Durham University Collections repository for the paper, https://doi.org/10.15128/r1d504rk42h.

## Code availability

The code used to process the fluorescence images and produce the figures in the paper is also available in the same repository, https://doi.org/10.15128/r1d504rk42h.

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

## Acknowledgements
We thank Tim de Jongh for valuable discussions on fluorescence imaging of atoms, Erkan Nurdun for the development of the moving dipole trap and Antonio Rubio-Abadal for sharing data to help the development of our image processing code. We also thank Daniel Ruttley and Caleb Rich for comments on the manuscript and presentation of the results. We acknowledge support from the UK Engineering and Physical Sciences Research Council (EPSRC) Grants EP/P01058X/1 and EP/W00299X/1, UK Research and Innovation (UKRI) Frontier Research Grant EP/X023354/1, the Royal Society, and Durham University. PDG is supported by a Royal Society University Research Fellowship URF/R1/231274 and Royal Society research grant RG/R1/241149.

## Author contributions
J.M.M., B.P.M., and A.P.R. performed the experiments. J.M.M. analysed the data with assistance from B.P.M. and A.P.R., P.D.G., and S.L.C.; J.M.M. wrote the original draft of the paper, and all authors reviewed and edited it; S.L.C. supervised the work and managed funding acquisition.

## Competing interests
The authors declare no competing interests.
