## [Transparent Peer Review file · Nature Communications]

Multi-state detection and spatial addressing in a microscope for ultracold molecules

Corresponding Author: Dr Jonathan Mortlock

Version 0:

Reviewer comments:

Reviewer #1

(Remarks to the Author)

This work reports two important advances in manipulating and detecting ultracold molecules. To date, ultracold molecular gases produced by assembly from degenerate atom gases have mostly been probed at a coarse level with absorption imaging. A few years ago, a group introduced quantum gas microscopy for detecting individual NaRb molecules in an optical lattice. The RbCs microscope described in this work is only the second such realization, but this paper goes beyond just imaging the molecules with single-site resolution. First, they encode a spin-1/2 in two rotational states of the molecule and show they can read out the spin state with single-site resolution. This is done by mapping the rotational state to whether they have a Rb or Cs atom on a site before imaging. This readout is crucial for studying quantum spin systems of molecules, especially when the lattice is far from unit filling (as is the case for all lattice experiments to date). Second, they demonstrate the ability to locally address the molecules and flip part of the cloud to a different spin state. These are important advances that will have a strong impact on the field of ultracold molecules and AMO more generally. The paper is well-written and the relevant work is properly cited. I recommend publication in this journal.

I have two suggestions to improve the work. First, one difference from the previous molecule microscopy work is that the molecules are produced in the bulk rather than in a lattice, and the preparation scheme means that the molecules are not in the ground state of the lattice when pinned. It would be interesting to report the average number of vibrational quanta occupied in each direction, and to discuss how this would impact the feasibility of studying quantum magnetism with the system. Second, while I appreciate that this is not a quantum computing type paper, I think the authors can still be a bit more quantitative in some parts of the paper. Both of the key results (state detection and local manipulation) do not have any fidelity associated with them as far as I can tell.

I believe there might be a typo in Fig. 3a: for the $N=1$ state (top row), shouldn't the state of the Rb atoms after dissociating the molecule be (1,1) rather than (2,2)? (after the ARP, they become (2,2)).

Reviewer #2

(Remarks to the Author)

The manuscript by Mortlock et al. describes the ability to freeze Rb or Cs atoms and/or RbCs molecules in a deep optical lattice and perform dual-species quantum gas microscopy on them. These techniques offer a versatile platform for studying atomic and molecular systems and mixtures in bulk or in an optical lattice. The authors exploit these abilities to study the microscopic distribution of molecules and to study the lifetime of a mixture of molecules and atoms.

Most importantly, the authors use these techniques to perform state-resolved detection of molecules. Since the occupation of molecules is not known because the creation efficiency of molecules is limited or because of any molecular loss processes, it is generally challenging to fully characterize a molecular system when two rotational "spin" states are used. Crucially, the spin state is detected by mapping it onto occupation. However, this mapping is flawed when occupation is a priori unknown. Therefore, it is crucial in molecular systems (and also recently becoming crucial in atomic quantum computers) to independently measure occupation and spin. The authors present a beautiful demonstration of this technique. The technique, as proposed back in 2018, is based on uniquely mapping each spin state to an atomic species. By independently detecting both atomic species, it is possible to detect both the occupation and the spin state.

Finally, the authors combine this state-resolved detection tool with local optical control to create a spin pattern in the ensemble. They use a local light shift on the rotational transition such that the light shifted region is off-resonant with the global microwave pulse that flips the atomic spin. They use this ability to demonstrate the power of their microscopic state-resolved detection. By switching between bulk ensembles and pinned molecules in a lattice, they demonstrate some preliminary dynamical behavior. However, the true power of these techniques is in the context of dipolar spin-exchange interactions and interacting spin dynamics, which presumably the authors leave for future work.

The paper is well written, the figures are clear, and the authors have thoroughly investigated the dual species microscopy of the Rb/Cs/RbCs system. I believe that the paper is suitable for publication in Nature Communications. I have some questions and comments that the authors should consider upon resubmission.

1. For the Rb+RbCs mixture lifetime, is it possible to detect RbCs using Cs atoms, thereby uniquely mapping either Rb atoms or molecules to an atomic species? Presumably, this would require a spin-flip of Rb right before the STIRAP back, such that atomic Rb can be differentiated from the molecular Rb, and the latter can be removed with a blast pulse. If so, it should be possible to obtain single-shot "species-resolved" images of the atom-molecule mixture. Do I understand correctly that this was not demonstrated here?
2. For the two-state detection, the authors should elaborate on what additional information the combined Rb-Cs image provides compared to the Rb-only and Cs-only images. Certainly, it provides information about loss or tunneling probability during detection. Perhaps this could be quantified by a subtraction or correlation plot.
3. As a related point, it would be interesting to see correlation plots between Rb and Cs events in their respective single-species images. Presumably, events in which a Rb and a Cs are on the detected on the same site are extremely rare due to light assisted collisions, as the authors mention. But I imagine that it could be possible to get a positive detection of the illuminated species in some shots before the pair is lost.
4. It may be helpful for the authors to briefly describe the reason why they need to ramp to low field for the Rb ARP. The readers recognize that this is a technical issue that will be solved, but the current situation is a curiosity that may be worth mentioning. Presumably it is related to the resonant bandwidth of the microwave coil or similar.
5. It may be appropriate to quantify the effect of the STIRAP light on the N=1 "up spin" molecules. Certainly, the effect would be small, but the statement could be more precise than "remain unaffected". Assuming the STIRAP and detection efficiency may be limited to ~0.99, we might assume that this is a good baseline for any deleterious effect on "up" molecules during the STIRAP step of "down" molecules. This means that the lifetime of "up" molecules while the STIRAP light is on should be ~100x longer than the STIRAP pulse time. The authors should consider measuring the lifetime of "up" molecules while the STIRAP light is on, where presumably the down-leg would be the source of any problem.

Version 1:

Reviewer comments:

Reviewer #1

(Remarks to the Author)

I am satisfied with the responses of the authors to my questions and the changes made to the manuscript.

Reviewer #2

(Remarks to the Author)

The authors have carefully and thoroughly addressed the comments raised by both referees and have taken more data that substantially strengthens the manuscript. They have studied and documented the mapping process fidelity in great detail, following two procedures that each have pros and cons with respect to the authors' ability to perform a microwave ARP at high field. Additionally, they have taken data to carefully measure the loss probability of "up" molecules during the STIRAP transfer process for the "down molecules". With these additions to the manuscript, I have no further reservations and suggest publication.

Response to the reviewers' comments

We thank both reviewers for their time and thoughtful comments on the paper. We are pleased that the reviewers both recommend publication in *Nature Communications*.

Both reviewers raise a small number of minor points that we address below. We are grateful for their suggestions which we believe have improved the clarity of the manuscript considerably.

Reviewer 1. Had two suggestions to improve the work.

1. “*First, one difference from the previous molecule microscopy work is that the molecules are produced in the bulk rather than in a lattice, and the preparation scheme means that the molecules are not in the ground state of the lattice when pinned. It would be interesting to report the average number of vibrational quanta occupied in each direction, and to discuss how this would impact the feasibility of studying quantum magnetism with the system.*”

We thank the reviewer for this question. Below we have done the requested calculation and rewritten the final paragraph of the Discussion to clarify how our work fits into efforts to study quantum magnetism with highly controlled arrays of molecules.

We note first that what we present here is *not* the correct approach to study spin dynamics in a lattice. Our work is specifically targeting the ability to do spin-resolved single-molecule detection in bulk gases. This is a critical tool to address open questions in the field. To study spin dynamics, one would follow the procedure used by Bakr/Nagerl and prepare the molecules in *ground band* of an optical lattice. This work is ongoing in our lab and we highlight this in the outlook of the paper.

However, for completeness we have analysed the prospect of spin dynamics in the current setup as requested. Details are given below:

From the temperature we observe in Fig. 4e, and our estimates of the trapping frequency after pinning (~60kHz radial, ~5kHz axial) we estimate mean motional quantum numbers after pinning of 1.3 radially and 10 axially. We identify two effects that motional excitation would have on studying quantum magnetism using dipole-dipole interactions between pinned molecules. 1. Differential light shifts 2. Variation in dipole-dipole interaction strength from the $\frac{1}{r^3}$ dependence.

In the current setup we expect differential light shifts will dominate. We can estimate this with a simple model: assuming that the trapping potential is from a harmonic intensity profile, $I = I_0(1 - x^2)$, we can see that for a given harmonic oscillator state the expectation value of the intensity goes as $I_n = I_0(1 - \frac{\hbar}{2m\omega}(2n + 1))$. Further algebra

allows us to derive the AC stark shift for a given harmonic oscillator state, n , as $\hbar\delta_n = \frac{\Delta\alpha}{2\alpha}\hbar\omega(2n + 1)$, with $\Delta\alpha$ being the difference between the polarizability of the two rotational states.

For the situation in this work, $\frac{\Delta\alpha}{\alpha} \sim 0.2$ and so the differential light shifts are on the scale of the trapping frequency and thus much stronger than our expected dipole-dipole interactions on the order of 100Hz, such that the interactions will be far off resonant for most pairs of molecules.

In future work we could change the trapping light to be at the magic wavelength for RbCs, which should eliminate this problem, leaving only spin-motion coupling from the radial $1/r^3$ dependence of the dipole-dipole interaction. Preliminary calculations indicate that for the 5uK temperature we measure we should expect significant broadening of the dipole-dipole coupling strength ($\frac{\Delta J}{J} > 0.1$), mainly due to the weak axial confinement.

To make it more clear how our current work relates to quantum magnetism we have reworded the final discussion paragraph, see page 7 and 8 of the resubmitted paper.

- 2. Second, while I appreciate that this is not a quantum computing type paper, I think the authors can still be a bit more quantitative in some parts of the paper. Both of the key results (state detection and local manipulation) do not have any fidelity associated with them as far as I can tell.*

We agree with the reviewer that although not essential to the paper, quoting fidelities in the main text would be interesting to some readers. We have therefore quantified the fidelity of each stage of the spin-mapping protocols.

We have investigated two experimental approaches to spin-mapping, both of which are now presented in the resubmission.

Sequence A: Turning off the high field coils for ARP, at the expense of a longer hold of the N=1 molecules in the lattice

Sequence B: Performing ARP at the STIRAP field to minimise the hold of N=1 molecules at the expense of ARP fidelity due to increased magnetic field noise.

The breakdown of the fidelities associated with each stage of these sequences are as follows:

- Rb ARP Fidelity
 - We measure a fidelity of Rb ARP transfer of 97(1)% in sequence A and 94(1)% sequence B.

- We note that N=0 detection relies on two ARP transfers, whereas N=1 detection only requires one ARP transfer.
- Survival of “N=1” molecules in the pinning lattice whilst ARP is performed
 - We measure the lifetime of the molecules after pinning, and from this estimate the loss for the hold times used in the two sequences. From these measurements we get the estimated survival probabilities of 86(10)% in sequence A (where the hold is 32ms) and 97(1)% in sequence B (where the hold is 8ms).
 - The atomic lifetime by contrast is much longer, on the order of seconds, and spontaneous scattering to other states is not problematic as all states are imaged by the optical molasses.
- Fidelity of STIRAP (95(1) % for both states)
 - In our experiment the STIRAP fidelity is limited by the amount of laser power available. We note that using higher laser power, our group has demonstrated 98.7(1) % transfer fidelity (Maddox et al. PRL 2024) and we are in the process of making this upgrade.
- Fidelity of Blasting of Rb and Cs atoms (>99%)
 - As mentioned in the relevant section of the methods, we observe a mean of <2 atoms remaining if we don't recover any molecules with STIRAP. These are leftover atoms from the >10,000 atoms used to make the molecules.
- N=1 -> N=0 Pi pulse fidelity (>99%)
 - Measured from the lowest number of detection events when we only recover N=0 after a pi pulse.

Combining all these factors together we estimate detection fidelities of:

Sequence A: 91(2) % for N=0 molecules and 78(9) % for N=1 molecules

Sequence B: 84(2) % for N=0 molecules and 84(2) % for N=1 molecules

We have updated the methods section to include a section *Fidelity of the spin-mapping process* (on page 9), detailing the above analysis, and made the description of the spin resolution in the results section generic (describing both A and B approaches). The data in Figure 3c has been replaced with data taken using method B.

Spatial addressing

The resolution of spatial addressing is limited by the spatial rate of change of the light shift at the intensity where the light shift and the Fourier width are nearly equivalent. Using the numbers reported in the paper, we calculate the addressing beam the gradient in the light-shift at the cross over point to be 3.2 kHz/ μm , which for the microwave pulse time used in our experiment gives a resolution of around 5 μm .

Ultimately this resolution is limited by the resolution of the imaging system. In future we plan to add in a SLM system to control addressing light projected through the microscope objective (rather than focused using a low NA lens above the cell). This will allow addressing potentials with significantly sharper edges and will enable a resolution approaching the lattice spacing. The spin resolution is also in theory limited by the fidelity of the pi pulse, which could be improved with composite pulse schemes.

We have added a comment on the resolution of spatial addressing in the results (on page 7).

We believe these changes help make the paper more quantitative, addressing the reviewer's concern.

I believe there might be a typo in Fig. 3a: for the $N=1$ state (top row), shouldn't the state of the Rb atoms after dissociating the molecule be (1,1) rather than (2,2)? (after the ARP, they become (2,2)).

We thank the reviewer for spotting this typo and have rectified the figure as required.

Reviewer 2. Had the following comments for us to consider

1. For the Rb+RbCs mixture lifetime, is it possible to detect RbCs using Cs atoms, thereby uniquely mapping either Rb atoms or molecules to an atomic species? Presumably, this would require a spin-flip of Rb right before the STIRAP back, such that atomic Rb can be differentiated from the molecular Rb, and the latter can be removed with a blast pulse. If so, it should be possible to obtain single-shot "species-resolved" images of the atom-molecule mixture. Do I understand correctly that this was not demonstrated here?

It is indeed possible to image Rb atoms and RbCs in the same sequence, using the protocol outlined by the reviewer, and we thank the referee for this suggestion.

The measurement we did was constructed as an example of measuring an impurity of RbCs in a bath of Rb, such that the loss of Rb atoms is expected to be small. We chose to measure the Rb density using absorption imaging as it was too high for single site resolved imaging (with $>10,000$ Rb atoms in the measurement, comparable to the number of sites in the lattice, we would expect very large parity projection losses).

Building on the reviewer's suggestion it would be possible to study a different system where the densities are comparable by reducing the Rb density, e.g. via a pulse which

transfers a finite, calibrated, fraction to the $F=2$ state followed by removal with $F=2 \rightarrow F'=3$ light as in the spin resolution sequence.

2. For the two-state detection, the authors should elaborate on what additional information the combined Rb-Cs image provides compared to the Rb-only and Cs-only images. Certainly, it provides information about loss or tunneling probability during detection. Perhaps this could be quantified by a subtraction or correlation plot.

We agree with the reviewer that the joint Rb-Cs image is not strictly needed to establish the molecular state distribution. It certainly contains information on loss of atoms and tunnelling. In practice, it was used to establish the relative positions of the lattice in the two images. Shifts between the images could arise from chromatic effects in the imaging system and/or drift of the lattice position over time. We see that the offset between the two lattices is smaller than one site, which makes it straightforward to assign the detected fluorescence to positions in the same underlying lattice structure.

To clarify this point, we have added the text in bold to the quoted section:

“Thus to reliably determine the relative locations of the atoms we take three successive images as shown in Fig. 3b collecting fluorescence first from Rb only, then Cs only, and finally Rb and Cs together. **The final image removes any ambiguity in the position of the lattice between different images.** This allows us to reconstruct a complete picture of the spin density as shown”

We also independently quantify the probability of loss or thermal hopping during imaging, as discussed in the methods.

3. As a related point, it would be interesting to see correlation plots between Rb and Cs events in their respective single-species images. Presumably, events in which a Rb and a Cs are on the detected on the same site are extremely rare due to light assisted collisions, as the authors mention. But I imagine that it could be possible to get a positive detection of the illuminated species in some shots before the pair is lost.

The reviewer is correct that events where a Rb and Cs are detected on the same site are very rare. Analysing spin-resolution data, we find that we have around 0.005 events per detected molecule where Rb and Cs are detected on the same site. This is despite the fact that the detection protocol begins by breaking apart the molecules to produce an initially highly correlated system with one Rb and one Cs atom in each site where there was a molecule.

We attribute the observation of a small but finite number of events where Rb and Cs are detected to the following possible experimental imperfections:

1. When a molecule is dissociated on a lattice site, there is an error in the ARP such that Rb isn't blasted away as expected. This leaves an atom pair after spin mapping. The Cs atom survives for the duration of the Rb fluorescence imaging and fluoresces enough to be detected.
2. During imaging, a Cs atom hops onto the site occupied by a Rb atom such that it is later detected on that site in the Cs image.
3. The reconstruction algorithm incorrectly assigns a neighbouring site as occupied due to the finite size of the atomic point spread function. These false positives could lead to assigning a single site with two molecules.

As the reviewer notes, process 1. is suppressed by light assisted collisions (LACs) of atoms during fluorescence. It is worth noting, however, that we have observed that there can be a relatively long lifetime ($>100\text{ms}$) of Rb-Cs pairs against LACs when the imaging light for only one species is present, as is the case here. This is remarkably long compared to the homonuclear case.

For completeness we include below a plot of the correlations, expressed as the probability of observing a Rb atom given the observation of a Cs atom at a site offset by a vector of lattice sites, normalised to the total number of molecules.

The background probability here (around 0.025) indicates the likelihood that two molecules from the gas would be pinned to the same site. However, we expect these molecules would be rapidly lost with high probability *prior to detection* due to light-induced loss of the two-molecule collision complex, as observed in many other ultracold molecule experiments.

4. It may be helpful for the authors to briefly describe the reason why they need to ramp to low field for the Rb ARP. The readers recognizes that this is a technical issue that will

be solved, but the current situation is a curiosity that may be worth mentioning. Presumably it is related to the resonant bandwidth of the microwave coil or similar.

The reviewer is correct that this was linked to the resonant bandwidth of the microwave antenna. Since our first submission we have replaced this antenna, and worked to reduce high frequency magnetic field noise, such that we can drive the MW transition for ARP near the magnetic field required for STIRAP, as detailed in our response to reviewer 1. This allows us to significantly reduce the time required for ARP, leading to much closer agreement in the total molecule number as measured by the two atoms.

The data in Figure 3c has been updated with this method, along with the corresponding discussions in the Results and Methods sections.

The new figure is copied below for convenience.

5. It may be appropriate to quantify the effect of the STIRAP light on the N=1 “up spin” molecules. Certainly, the effect would be small, but the statement could be more precise than “remain unaffected”. Assuming the STIRAP and detection efficiency may be limited to ~0.99, we might assume that this is a good baseline for any deleterious effect on “up” molecules during the STIRAP step of “down” molecules. This means that the lifetime of “up” molecules while the STIRAP light is on should be ~100x longer than the STIRAP pulse time. The authors should consider measuring the lifetime of “up” molecules while the STIRAP light is on, where presumably the down-leg would be the source of any problem.

It is known that the Stokes light can affect the N=1 molecules, due to the presence of nearby excited states around 1GHz detuned. The spectroscopy of these states is

detailed in the theses of Markus Debatin (Innsbruck 2013) and Peter Molony (Durham 2016).

As suggested by the reviewer, we have now measured the probability of loss of N=1 molecules during the STIRAP pulse by comparing two sequences, one where we apply 10 consecutive STIRAP transfers (pulses which would transfer from N=0 to the Feshbach state), and a control measurement in a sequence which is identical but omits the STIRAP light. We found from this measurement that there is indeed measurable loss (3(1)%) of the N=1 molecules during the application of STIRAP to N=0 molecules, which we have now noted in the paper in the new section on the *Fidelity of the spin-mapping process* in the Methods on page 9.